# Manipulation with Mutational Status of VHL Regulates Hypoxic Metabolism and Pro-Angiogenic Phenotypes in ccRCC Caki-1 Cells

**DOI:** 10.3390/ijms262110629

**Published:** 2025-10-31

**Authors:** Pavel Abramov, Alexandr Mazur, Aleksey Starshin, Svetlana Zhenilo, Egor Prokhortchouk

**Affiliations:** 1Federal Research Centre «Fundamentals of Biotechnology», Russian Academy of Sciences, 119071 Moscow, Russiastarshin.alexey@gmail.com (A.S.); prokhortchouk@gmail.com (E.P.); 2Institute of Gene Biology, Russian Academy of Sciences, 119334 Moscow, Russia; 3Group of Epigenetic Editing, Research Center for Genetics and Life Sciences, Sirius University of Science and Technology, 354340 Sirius, Russia

**Keywords:** VHL, hypoxia, renal cancer, single-cell

## Abstract

Clear cell renal cell carcinoma (ccRCC), accounting for 80–90% of renal malignancies, is frequently driven by VHL inactivation—either through mutation or promoter hypermethylation—resulting in constitutive HIF2α activation and pseudohypoxic signaling. VHL gene inactivation is a hallmark of von Hippel–Lindau syndrome, a hereditary disorder predisposing patients to ccRCC and other tumors, underscoring its central role in disease pathogenesis. While VHL dysfunction promotes aggressive tumor phenotypes, the therapeutic potential of VHL restoration remains underexplored. Here, using the Cas9 induced VHL-mutation in the Caki-1 cell line model, we demonstrate that VHL inactivation augments hypoxia-like pathways and enhances anaerobic glycolysis. Rescue of functional VHL reversed these activation patterns and modulated the expression of genes associated with angiogenesis. Using single cell transcriptomics, we show that the VHL-positive and -negative Caki-1 cells are characterized with different proportions of benign and aggressive cells as seen by analysis of specific gene expression. Furthermore, the identified angiogenesis-related genes were linked to affect clinical outcomes in ccRCC patients, suggesting that VHL restoration may mitigate high-risk molecular features.

## 1. Introduction

Kidney or renal cancer is one of the most common cancers in the United States. In 2024, there were 80,980 new diagnosed cases and 14,510 deaths from kidney cancer [1]. In Russia, during the period from 2012 to 2017, the standardized incidence rates increased by 13.8% among men (from 12.3 to 14.0 per 100,000 population, world standard) and by 16.4% among women (6.7 and 7.8 per 100,000 population) with a total of 24,800 new diagnosed cases of renal cancer [2]. In the world, approximately 80% of renal carcinomas are clear cell tumors (ccRCC). Cells of origin in ccRCC have been identified as proximal tubular epithelial cells and are more likely to hematogenously metastasize to the lungs, liver, and bones [3]. The inactivation of somatic mutations or deletions in the von Hippel–Lindau (VHL) tumor suppressor gene region (chromosome arm 3p) is strongly associated with bilateral manifestation of younger age (20–40 years) [3]. In non-familiar ccRCC cases, somatic *VHL* mutations are found in 90% of cases [4].

The VHL gene serves as a key tumor suppressor gene preventing cell proliferation, angiogenesis, and cell differentiation. It functions as a key regulator of hypoxia. It directs poly-ubiquitination of HIF1-a and induces its degradation. This leads to negative regulation of hypoxia-associated genes [5]. *VHL* mutations disturbs this regulation axis and leads to consistent activation of the HIF pathway. This leads to a pro-hypoxic tumor environment. Such tumors have more aggressive phenotype and poor prognosis [6]. One of the factors contributing to the metastatic potential of renal cancer cells is periostin, a matricellular protein secreted by VHL-deficient kidney cells. Periostin facilitates the migration and dissemination of VHL-positive cells to distant tissues, thereby promoting metastatic progression [7].

According to the disruption of *VHL* function in the vast majority of ccRCCs, gene therapy looks promising. It involves replacing or repairing the mutated *VHL* gene in cells affected by the disease [8]. For example, ultrasound microbubble-mediated *VHL* expression in the OVCAR3 cell line inhibited proliferation and migration, caused cell-cycle arrest, and promoted apoptosis [9]. Moreover, polyethyleneimine-derived nanoparticles with *VHL* plasmids successfully reduced tumor volume in BALB/c nude mice with established renal cell carcinoma [10]. Studies using the Caki-1 cell line, a human ccRCC model, demonstrate that restoration of *VHL* function after mutation leads to a decrease in tumorigenicity, which can also be considered a potential therapeutic approach [11]. However, a critical gap remains in understanding how restoration of *VHL* impacts heterogeneous tumor subpopulations.

While bulk RNA sequencing captures broad transcriptional changes, it obscures cell-to-cell variability, masking nuanced responses to therapy. However, single-cell RNA sequencing (scRNA-seq) offers unparalleled resolution, enabling the study of gene expression patterns at the individual cell level. A key advantage of scRNA-seq is the ability to analyze pathway activity across distinct cell populations using methods like Gene Set Variation Analysis (GSVA) [12]—a powerful approach for evaluating the enrichment of predefined gene sets in single cells without requiring predefined groups. With this tool, we can quantify pathway-level activity in individual cells, uncovering how key biological processes such as glycolysis, hypoxia response, or epithelial mesenchymal transition (EMT) are dynamically regulated within tumor subpopulations. This method is particularly powerful in cancer research, where metabolic reprogramming and signaling pathway crosstalk often drive progression and therapy resistance.

In this study, we investigated how cellular heterogeneity changes in model cell lines upon *VHL* mutation and following its restoration, using single-cell transcriptome analysis. As a model system, we chose the Caki-1 cell-based model previously described and characterized by our group [13]. In this model, *VHL* inactivation via CRISPR-Cas9 resulted in HIF-1α protein accumulation under normoxic conditions similar to the levels of hypoxia-cultivated cells [14]. Additionally, clones with mutated *VHL* gave rise to significantly larger tumors in immunodeficient mice as compared to the original Caki-1 cells with wild-type *VHL*. Importantly, restoring functional *VHL* in cells significantly reduced tumor growth compared to the mutant *VHL* cells [11]. Given this established link between *VHL* status and tumor aggressiveness, we now utilize single-cell RNA sequencing to deconvolve the cellular heterogeneity within this model.

In this work, we demonstrated that scRNA-seq experiments reproduce bulk RNA sequencing conclusions that *VHL* mutation produces hypoxia-like expression patterns. Bwith error(s) ased on single-cell transcriptomic data, the Caki-1 cell population can be divided into two distinct subsets. The gene expression profiles of these subsets indicate that they exhibit different malignancy phenotypes: one that is pro-angiogenic and one that is normal. These observations can be made in all cell lines despite *VHL* status.

## 2. Results

### 2.1. Integrated Single-Cell and Bulk Transcriptomic Analysis Reveals VHL-Dependent Gene Expression Programs in Caki-1 Cells

We performed single-cell transcriptomic profiling on four isogenic Caki-1 cell lines: the parental wild-type cell line (WT), a VHL-mutant cell line (mutVHL) with a CRISPR/Cas9-induced frameshift mutation in the C-terminal α-domain (Appendix A), a mutVHL cell line with restored *VHL* (mutVHL-restored), and a mutVHL cell line with empty vector (mutVHL-control). Restoration of functional *VHL* expression in the mutant cells was achieved via lentiviral transduction with a VHL-HA expression construct (Appendix A), and mutVHL-control cells were transduced with an empty control lentiviral vector [11]. After quality control of single cell RNA sequencing data, we analyzed RNA expression in 4303 WT, 4341 mutVHL, 4312 mutVHL-restored, and 4992 mutVHL-control cells.

Initially, we compared the single-cell transcriptomic data with the bulk RNA-sequencing data. In our previous work with bulk RNA-seq, we detected that *VHL* inactivation led to a significant increase in gene expression of the enzymes involved in all stages of glycolysis, including *SLC2A1*, *HK1*, *PFKL*, *PFKP*, *ALDOA*, *ALDOC*, *GAPDH*, *PGK1*, *PGAM1*, *ENO1*, *ENO2*, and *LDHA* [11]. As shown in Figure 1A, the normalized expression levels in WT, mutVHL, and mutVHL-restored cell lines show strong correlations across both bulk and single-cell RNA-seq datasets. In all comparisons, whether between different cell lines (WT, mutVHL, or mutVHL-restored) or between bulk and single-cell RNA-seq methods, we consistently observed at least moderate positive correlations in normalized gene expression levels. According to this, single cell and bulk RNA-seq experiments show similar results, and we can extrapolate previous findings on the single cell data.

Due to the highly correlated RNA expression levels observed in our single-cell experiment, we were able to separate the cells only into two groups using the Leiden clustering algorithm: one cluster is dominated by cells with functional *VHL* and another by cells with mutant *VHL* (Figure 1B,C). The top 15 differentially expressed genes (DEGs) are shown in Figure 1D and include genes involved in glycolytic processes, such as *PGK1* (LogFC = −1.48, p_adj = $7.14\times 10^{-293}$), *LDHA* (LogFC = −1.36, p_adj = $7.14\times 10^{-293}$), *TPI1* (LogFC = −0.89, p_adj = $7.14\times 10^{-293}$), and *PKM* (LogFC = −0.89, p_adj = $7.14\times 10^{-293}$). Notably, *LDHA* and *BNIP3* are directly regulated by *HIF1-*α, as previously described [15]. The complete set of DEGs is provided in Appendix A.

We found 4504 DEGs between normal and mutVHL cells in our single cell experiment. While 2523 of the identified DEGs overlap with DEGs from our previous bulk RNA study, 6424 DEGs were found uniquely in the bulk experiment (Appendix A). This effect is likely caused by technical limitations of single-cell library preparation and the loss of some low-abundance genes detected in bulk RNA sequencing. This suggests that, while bulk RNA-seq captures a broader dynamic range of gene expression, single-cell RNA-seq reliably recapitulates major transcriptional differences between VHL-functional and VHL-mutant states.

### 2.2. Gene Set Variational Analysis Shows Subsets of Cells with a Different Malignancy Phenotype

Comparing pathway activation via GSVA scores in WT cells with other cell lines at single-cell resolution could reveal how specific metabolic or survival pathways contribute to aggressive phenotypes.

For the GSVA analysis, we selected hallmark gene sets from the Human Molecular Signatures Database (MSigDB) Collections [16,17]. These hallmark gene sets represent well-defined biological processes and are widely used to summarize pathway activities in transcriptomic studies. We compared the enrichment of these gene sets between WT cells and every other experimental group of cells. The top 8 differentially enriched pathways are presented in Table 1 and Figure 2. A full list of comparisons is available in Appendix A. Compared to WT cells, the most differentially enriched pathways in both mutVHL and mutVHL-control cells were Hypoxia, Glycolysis, mTORC1, PI3K/AKT/mTOR, and P53 signaling pathways. The first two pathways are activated upon VHL inactivation and include genes that are tightly regulated by the VHL-HIF1α axis, such as *PGK1*, *PDK1*, *ALDOA*, *ALDOC*, *AKAP12*, *ANGPTL4*, *ANKZF1*, *ADM*, *VEGFA*, *SLC2A1*, and *BNIP3L* [15,18]. mTORC1, PI3K/AKT/mTOR, and P53 pathways are key regulators of cell growth, metabolism, and proliferation [19,20]. Compared to Caki-1 cells, cells with recovered *VHL* are enriched in WNT/β-Catenin, Inflammatory, and Angiogenesis pathways.

Based on the activity profiles of key signaling pathways-including Hypoxia, Glycolysis, mTORC1, PI3K/AKT/mTOR, P53, WNT/β-Catenin, Inflammatory, and Angiogenesis, we performed unsupervised classification of all cells using Gaussian mixture modeling. This analysis segregated the cells into two distinct phenotypic clusters, designated as “normal” and “pro-angiogenic,” with the Angiogenesis pathway serving as the principal discriminator between these groups (Figure 2C).

Quantitative assessment (Figure 2B, Table 2) revealed a marked enrichment of the pro-angiogenic phenotype within the mutVHL, mutVHL-control, and mutVHL-restored cells, exhibiting normal-to-pro-angiogenic cell ratios of 0.83, 0.87, and 0.93, respectively. In contrast, the WT cell line demonstrated a predominance of normal phenotype cells, reflected by a ratio of 1.59. These data indicate that *VHL* inactivation correlates with an increased proportion of cells exhibiting pro-angiogenic transcriptional programs, whereas Caki-1 cells maintain a lower frequency of such cells. Notably, cells with restored *VHL* expression (mutVHL-restored) displayed an approximately equal distribution between normal and pro-angiogenic phenotypes, suggesting only partial phenotypic reversion upon *VHL* restoration.

These findings demonstrate that, while *VHL* status significantly influences angiogenic signaling and cellular heterogeneity in the Caki-1 model system, it is not the sole determinant of these processes.

### 2.3. Comparison of Pro-Angiogenic and Normal Cells Gene Expressions

Next, we performed a comparative analysis of pro-angiogenic and normal cell subpopulations (Appendix A). In each cell line, the top five differentially expressed genes (DEGs) were *CCND2* (LogFC = 1.03, p_adj = $7.14\times 10^{-293}$), *TIMP1* (LogFC = 0.79, p_adj = $7.14\times 10^{-293}$), *LRPAP1* (LogFC = 0.66, p_adj = $4.93\times 10^{-197}$), *APP* (LogFC = 0.93, p_adj = $3.87\times 10^{-199}$), and *MSX1* (LogFC = 1.1, p_adj = $1.48\times 10^{-156}$) (Figure 3A). All of them are upregulated in pro-angiogenic cells. *CCND2* is one of the three D-cyclin genes. This gene is upregulated in multiple myeloma [21]. *TIMP1* encodes matrix metalloproteinase and is involved in EMT [22]. *LRPAP1* is shown as a key player in the micropapillary pattern metastasis of lung adenocarcinoma [23]. *APP* is also shown to be involved in proliferation and migration of cancer cells [24]. In contrast, *MSX1* exerts tumor-suppressive functions [25,26].

Next, we compared the subpopulations of all cell lines to the normal cell subpopulation of WT cells (Appendix A). Log2 fold-change values exhibited strong correlations across all subpopulations within mutVHL and mutVHL-control cell lines. Pro-angiogenic mutVHL-restored cells demonstrated a strong correlation with all mutVHL cell subpopulations, a moderate correlation with normal subtype of mutVHL-restored cells, and no correlation with WT pro-angiogenic cells (Figure 3B). This indicates that rescue of *VHL* expression alone is insufficient to fully reestablish the gene expression profile in Caki-1 cells.

Comparison of gene expressions in normal subpopulation of WT cells with pro-angiogenesis mutVHL-restored and normal mutVHL-restored revealed that *NDUFB10* (normal mutVHL-restored: LogFC = 0.5, p_adj = $2.1\times 10^{-165}$; pro-angiogenic mutVHL-restored: LogFC = 0.55, p_adj = $2\times 10^{-177}$) and *SNHG9* (normal mutVHL-restored: LogFC = 1.06, p_adj = $1.5\times 10^{-148}$; pro-angiogenic mutVHL-restored: LogFC = 1.07, p_adj = $3.7\times 10^{-158}$) genes were upregulated in both mutVHL-restored cell subpopulations. *NDUFB10* encodes a subunit of mitochondrial electron transport chain, and its knockout leads to impaired respiration and reduction of mitochondrial membrane potential [27]. *SNHG9* exhibits a dual role in cancer progression: it has been shown to promote cell proliferation, migration, and invasion in hepatocellular carcinoma cells [28], induce hepatoblastoma tumorigenesis via miR-23a-5p/WNT3a Axis [29], and be associated with with poor survival and immune infiltrations in prostate cancer [30]; in contrast it inhibits ovarian cancer progression by sponging microRNA-214-5p [31]. *VIM* is downregulated in both normal (LogFC = −0.68, p_adj = $5.2\times 10^{-162}$) and pro-angiogenic (LogFC = −0.58, p_adj = $3.4\times 10^{-134}$) mutVHL-restored cells compared to normal WT cells. Upregulation of *VIM* in ccRCC is a common occurrence in patients with mutated *VHL* [32]. *VIM* encodes a cytoskeletal protein, vimentin filament that supports mechanical integrity of the migratory machinery of a cell and is involved in the key events during EMT [33]. The *NUCKS1* gene is also downregulated in both mutVHL-restored cell subpopulations (normal mutVHL-restored: LogFC = −0.45, p_adj = $4.2\times 10^{-121}$; pro-angiogenic mutVHL-restored: LogFC = −0.48, p_adj = $8.5\times 10^{-153}$). However, this gene is associated with tumor proliferation, invasion, and progression [34,35].

All cell subtypes with mutated *VHL* showed upregulation in genes associated with glycolytic processes and hypoxia (Figure 3A).

## 3. Discussion

Our study reveals critical insights into the heterogeneity of Caki-1 cells and the functional consequences of *VHL* restoration. A key finding is the presence of pro-angiogenic cells across all experimental groups, including WT, mutVHL, and mutVHL-restored cell lines. This suggests that pro-angiogenic potential is an intrinsic feature of Caki-1 cells, though its prevalence is significantly influenced by *VHL* status. Notably, *VHL* restoration only partially normalized the ratio of normal to pro-angiogenic cells, indicating that functional *VHL* alone is not sufficient to fully revert the tumorigenic phenotype.

The pro-angiogenic subpopulations in all cell types, whether mutVHL, mutVHL-restored, or even WT, shared a core set of DEGs. All of them were upregulated in pro-angiogenic cells compared to cells classified within the normal subpopulation. Cyclin D2 is a member of the D-type cyclins that plays a critical role in regulating cell cycle, cellular differentiation, and malignant transformation. *CCND2* is usually downregulated in ccRCC tissues compared to adjacent non-malignant kidney tissues due to aberrant promoter hypermethylation. This epigenetic silencing of *CCND2* is thought to contribute to tumorigenesis by disrupting normal cell cycle regulation, as *CCND2* normally promotes cell cycle progression from G1 to S phase [36]. Nevertheless, in pro-angiogenic clusters, regardless of *VHL* status, we detected an increased transcription of *CCND2*. A similar pattern has also been observed in glioma and melanoma [21,37]. *TIMP1* (TIMP metallopeptidase inhibitor 1) is also upregulated in cells with the pro-angiogenic phenotype. It naturally inhibits matrix metalloproteinases and participates in extracellular matrix remodeling. Upregulating *TIMP1* accelerated the proliferation, migration, and invasion of RCC cells [38]. Dysregulation of *TIMP1* is associated with cancer progression and the accumulation of cancer associated macrophages and poor prognosis in high-risk surgically resected melanoma patients [39,40,41]. *TIMP1* and *CCND2* upregulation in pro-angiogenic cells mirror the POSTN-driven phenotype, shown in [7], implicating extracellular matrix remodeling and cell-cycle dysregulation as conserved mediators of cooperative aggression.

*LRPAP1* and *APP* are also upregulated in pro-angiogenic cells and involved in cell migration and metastasis [23,24]. *MSX1* has a dualistic relation with cancer. On the one hand, it promotes cell proliferation and invasion in human colon cancer cells [42]. On the other hand, it promotes cell cycle arrest in different cycle stages and works as a tumor suppressor [25,26]. The persistence of these DEGs in VHL-rescued cells implies that pro-angiogenic programs are resilient to *VHL* reintroduction, likely due to epigenetic or microenvironmental stabilization.

It is important to note, however, that our model relies on a single clonal isolate, so some differences between WT and mutVHL cells may reflect pre-existing heterogeneity rather than direct *VHL* effects. Despite this limitation, the shared DEG signature among pro-angiogenic cells across all genotypes reveals stable molecular drivers of aggressiveness that endure despite *VHL* recovery.

In conclusion, our findings demonstrate that the pro-angiogenic phenotype in Caki-1 cells is not solely dependent on the canonical VHL-HIF hypoxia axis but is instead supported by a network of alternative pathways, including ECM remodeling (*TIMP1*, *LRPAP1*, *APP*) and cell-cycle dysregulation (*CCND2*). We reveal that, even within an isogenic cell line model, there is a stable transcriptional heterogeneity that leads to the formation of normal and pro-angiogenic cell phenotypes. While *VHL* mutational status serves as a powerful modulator of the proportion of these aggressive pro-angiogenic cells, it is not an absolute prerequisite for their emergence. This underscores the complexity of tumorigenesis, where genetic drivers act upon a pre-existing landscape of cellular plasticity. Still it is unclear what kind of interactions between these pro-angiogenic and more normal cell subtypes acquire within the tumor microenvironment.

## 4. Materials and Methods

### 4.1. Cell Culture

The Caki-1 cell line was provided by Dr. Deyev Igor from the Institute of Bioorganic Chemistry. Deyev Igor bought them from ATCC (ATCC number HTB-46). WT (Caki-1), mutVHL, mutVHL-restored, and mutVHL-control cell lines [11] were maintained in Dulbecco’s Modified Eagle Medium (DMEM) supplemented with 10% fetal bovine serum (FBS), 1% penicillin/streptomycin, and 2 mM L-glutamine. The VHL-mutant cell line was generated via CRISPR-Cas9-mediated knockout (targeting exon 3) and validated by Sanger sequencing. To produce a VHL-mutant Caki-1 cell line, we performed a CRISPR-Cas9 genome editing. The guide RNA (gRNA) targeting exon 3 of the VHL gene (ENSG00000134086) was designed using CRISPR Design Tool (https://www.addgene.org/genome-engineering/, accessed on 16 September 2025) to maximize on-target efficiency and minimize off-target effects. The gRNA sequence (5′-AGGTCGCTCTACGAAGATC-3′) was cloned into the pSPCas9(BB)-2A-Puro (PX459) vector (Addgen) treated by the restriction endonuclease BbsI. The px459-gRNA(VHL) plasmid was then transfected into Caki-1 cells according to the (Ca^2+^)-phosphate transfection procedure (Promega, Madison, WI, USA). Following puromycin selection (puromycin-containing DMEM 10% FBS (2 mM), 48 h post-transfection), single-cell clones were isolated by limiting dilution (100 cells/10 mL). Genomic DNA was purified from expanded clones (days 34–48). Successful knockout was confirmed by Sanger sequencing of PCR-amplified genomic DNA spanning the target site (primers: Fw 5′-CATCAGCATAACACACTGCCA-3′, Rev 5′-GGAACCAGTCCTGTATCTAGA-3′). The PCR product was cloned into the T-vector with an InsTA clone PCR Cloning kit (Thermo Scientific, Waltham, MA, USA) and subjected to Sanger sequencing (Appendix A).

Unlike the clonal mutVHL cell line, which was isolated by limiting dilution, mutVHL-restored and mutVHL-control cell lines are polyclonal populations. These populations were generated by transducing the parental mutVHL cells with either the VHL-HA rescue construct or the empty pCDF1-MCS2-EF1-Puro (Addgen) vector, followed by puromycin selection (puromycin-containing DMEM 10% FBS (2 mM), 48 h post-transfection) to eliminate untransduced cells.

### 4.2. Single-Cell RNA Sequencing Data Analysis

#### 4.2.1. Single-Cell RNA Sequencing (scRNA-seq) Data Processing

Single-cell RNA sequencing libraries were prepared for each cell line. For each replicate, single-cell gene expression libraries were generated and sequenced. The resulting FASTQ files were processed using Cell Ranger (v9.0.1, 10× Genomics) count function with default parameters against the human reference genome (GRCh38 reference genome and gencode v44 primary assembly annotation from the 10× Genomics website) to generate a feature-barcode matrix for each individual sample. Further analysis was made using the Scanpy toolkit (v1.11.1) [43]. Raw gene expression matrices were subjected to standard quality control (QC) filters: cells with fewer than 500 detected genes or >20% mitochondrial reads were excluded; genes expressed in fewer than 10 cells were removed. Counts were normalized to median library size and log1p-transformed.

#### 4.2.2. Feature Selection and Dimensionality Reduction

Highly variable genes (HVGs) were selected using the Pearson residuals method (n = 3000 genes) [44]. Principal component analysis (PCA) was performed on scaled residuals, and the top 30 principal components (PCs) were retained for downstream analysis. Nonlinear dimensionality reduction was achieved via UMAP [45], with initial positions derived from partition-based graph abstraction based on cell type [46]. Unsupervised clustering was performed with Leiden algorithm [47] with 2 iterations and 0.04 resolution.

#### 4.2.3. Differential Expression and Pathway Analysis

DEGs between clusters were identified using the Wilcoxon rank-sum test, with Benjamini–Hochberg correction for multiple testing (adjusted *p* < 0.05). GSVA was performed using the GSEApy package (v.1.1.5) [48]. Hallmark gene sets from MSigDB were scored per cell using log-normalized expression data.

#### 4.2.4. Code Availability

Code used for analysis is accessed 10 September 2025 and available at https://github.com/pav1201/scVHL.

## 5. Conclusions

Our single-cell transcriptomic analysis reveals that the aggressive phenotype driven by *VHL* inactivation is underpinned by a fundamental and persistent reorganization of cellular heterogeneity. We demonstrate that, regardless of *VHL* status, the Caki-1 population is structured into two distinct subpopulations: one with a pro-angiogenic phenotype and one with a more normal phenotype. While *VHL* mutation amplifies the hypoxic signature and increases the prevalence of the pro-angiogenic cells, the restoration of functional *VHL* is insufficient to fully reverse the proportions of these populations. The resilience of the pro-angiogenic phenotype, even after genetic correction, points to the activation of robust VHL-independent pathways such as ECM remodeling and cell-cycle dysregulation that sustain cellular aggressiveness. 

## Figures and Tables

**Figure 1 ijms-26-10629-f001:**
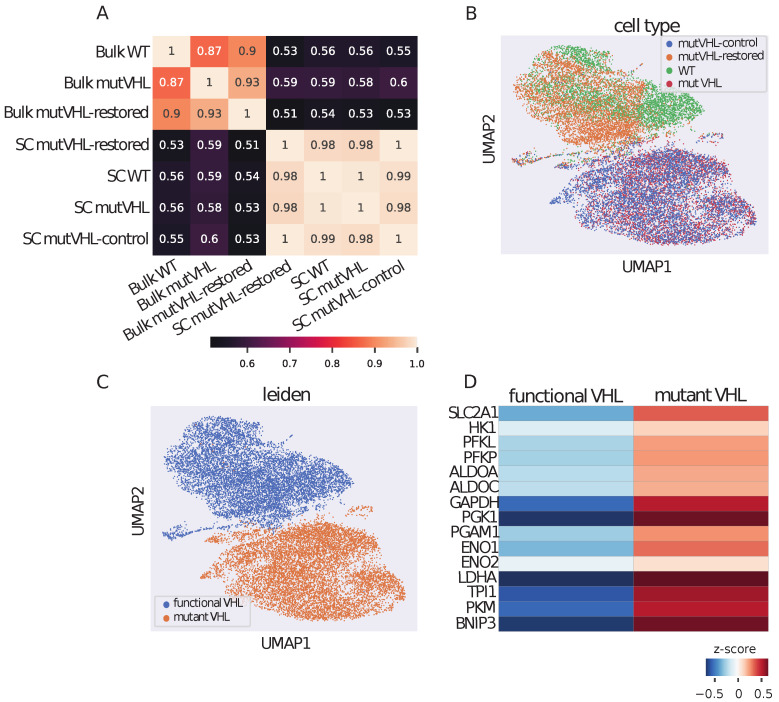
Single-cell transcriptomic profiling reveals genotype-driven clustering and metabolic reprogramming in VHL-engineered cell lines. (**A**) Correlation matrix of normalized RNA expression profiles across VHL genotypes in bulk and single-cell RNA-seq datasets. Pairwise Pearson correlations of gene expression levels (log-normalized counts) between WT, mutVHL, mutVHL-restored, and mutVHL-control cell lines are shown for both bulk and single-cell RNA sequencing experiments. (**B**) UMAP projections of single-cell data colored by cell lines. (**C**) UMAP projections colored by Leiden clusters, demonstrating genotype-dependent segregation. Functional VHL clusters (blue) are predominantly composed of WT and mutVHL-restored cells, while the VHL-mutant cluster (orange) is enriched for mutVHL and mutVHL-control cells, confirming successful genetic stratification. (**D**) Heatmap of mean z-scores of expression patterns of DEGs previously identified in bulk RNA-seq analyses ([11], highlighting conserved patterns of metabolic dysregulation).

**Figure 2 ijms-26-10629-f002:**
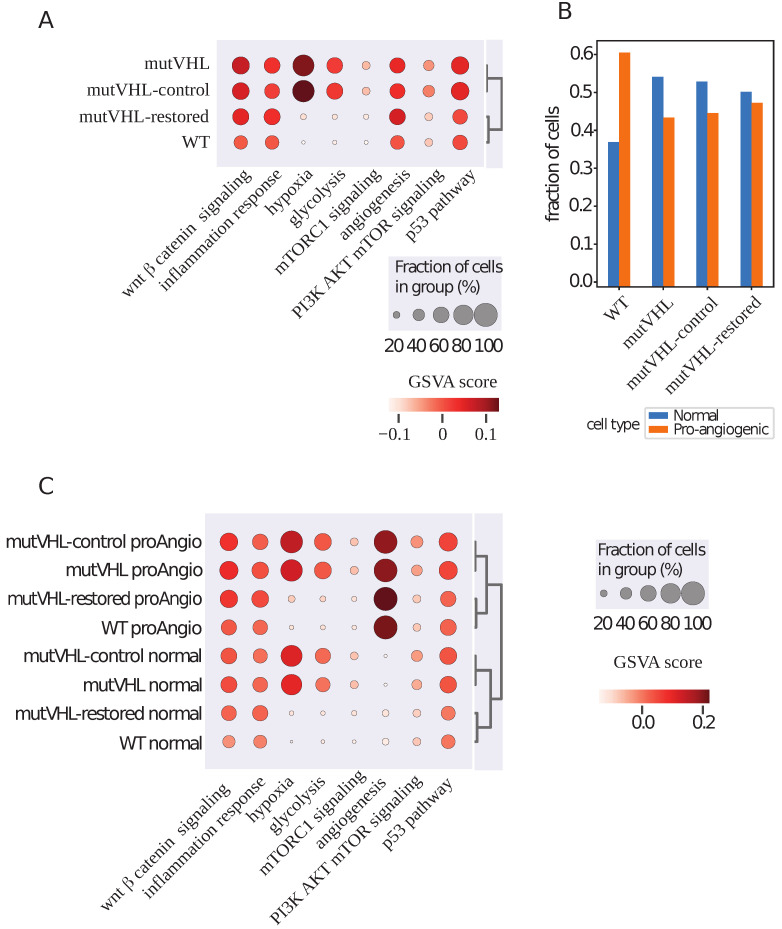
Pathway enrichment and pro-angiogenic subpopulation analysis in VHL-engineered cell lines. (**A**) Hallmark pathway enrichment analysis across cell lines. Dot plot shows enriched pathways analyzed by GSVA. Circle size represents the fraction of cells with pathway activity in each group (WT, mutVHL, mutVHL-restored, mutVHL-control); color intensity indicates mean enrichment score. Key pathways (e.g., hypoxia, glycolysis, angiogenesis) show genotype-specific activation patterns. (**B**) Cell fractions of normal and pro-angiogenic cellular subpopulations across experimental groups. Bar plot demonstrates the relative proportions of cells exhibiting pro-angiogenic signatures. (**C**) Subpopulation-specific Hallmark pathway enrichment. Dot plot displays pathway activation patterns within distinct cellular subpopulations (normal vs. pro-angiogenic) across genotypes. Circle size and color represent fraction of active cells and mean enrichment score, respectively, highlighting conserved pro-angiogenic pathway activation regardless of *VHL* status.

**Figure 3 ijms-26-10629-f003:**
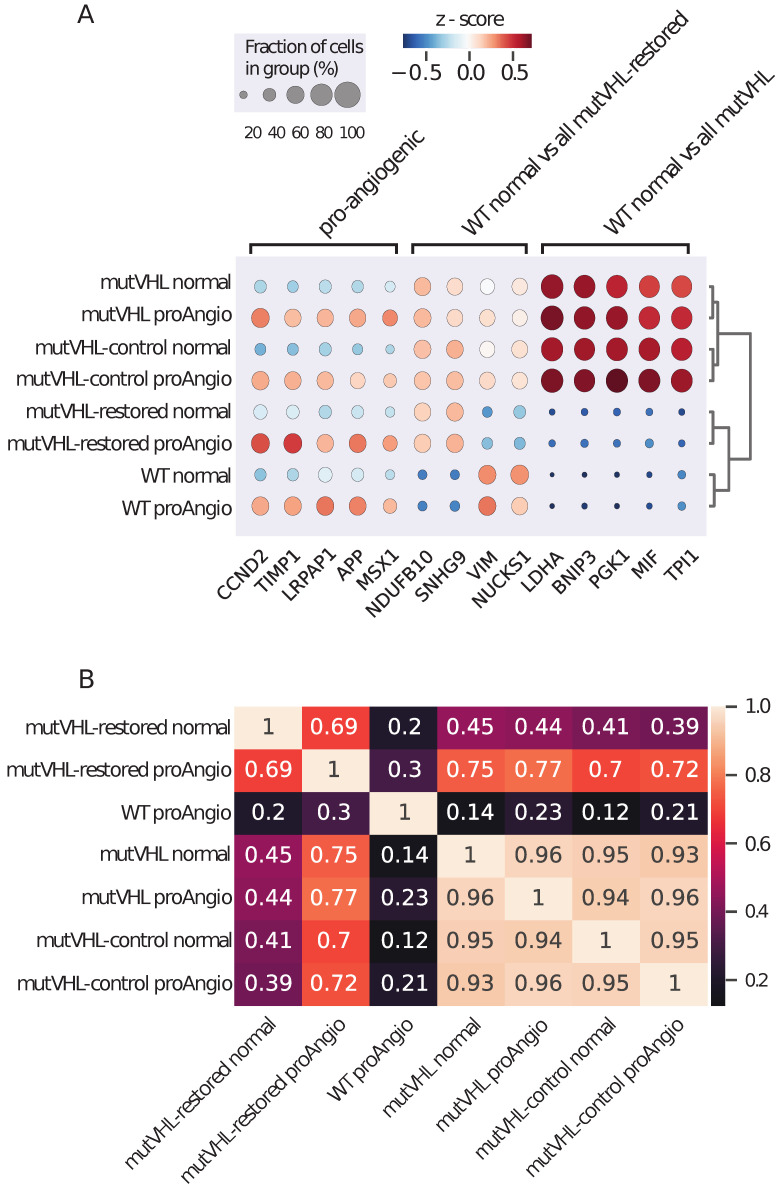
Transcriptomic and phenotypic characterization of pro-angiogenic subpopulations. (**A**) Dotplot of DEGs that were the result of comparison of normal and pro-angiogenic cell subpopulations (pro-angiogenic cluster), common DEGs from the comparison of WT normal cells with normal and pro-angiogenic mutVHL-restored cells (WT normal vs. all mutVHL-restored cluster) and common DEGs from comparison of WT normal cells with normal and pro-angiogenic cells from mutVHL and mutVHL-control cell lines. (**B**) Pairwise Pearson correlations of Log2FCs across all subpopulations of cell lines.

**Table 1 ijms-26-10629-t001:** Differential pathway enrichment between WT and other cell lines. Z-values from Wilcoxon rank-sum tests comparing the enriched pathways with the highest significance between WT and mutVHL, mutVHL-restored, and mutVHL-control cell lines. Pathways are ranked by the magnitude of differential enrichment (Z), highlighting persistent metabolic and angiogenic signatures despite VHL restoration.

mutVHL-Restored vs. WT Pathway Names	mutVHL-Restored vs. WT Z-Values	mutVHL vs. WT Pathway Names	mutVHL vs. WT Z-Values	mutVHL-Control vs. WT Pathway Names	mutVHL-Control vs. WT Z-Values
WNT-β-Catenin	14.74	Hypoxia	74.73	Hypoxia	78.52
Inflammatory response	13.59	Glycolysis	57.53	Glycolysis	60.81
Angiogenesis	12.81	MTORC1	25.12	MTORC1	24.63
TNFα via NFκB	12.31	WNT-β-Catenin	21.58	PI3K-AKT-MTOR	20.42
Hypoxia	12.20	P53	19.0	P53	19.89

**Table 2 ijms-26-10629-t002:** Counts of cells with normal and pro-angiogenic phenotype.

Cell Line	Normal Cell Counts	Pro-Angiogenic Cell Counts	Normal/Pro-Angiogenic Cell Counts Ratio
mutVHL-restored	2086	2226	0.93
WT	2644	1659	1.59
mutVHL	1973	2368	0.83
Ctr	2327	2665	0.87

## Data Availability

The raw data supporting the conclusions of this article will be made available by the authors on request.

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
