# Peer review of "Manipulation with Mutational Status of VHL Regulates Hypoxic Metabolism and Pro-Angiogenic Phenotypes in ccRCC Caki-1 Cells"

_ijms, 2025, doi:10.3390/ijms262110629_

Round 1
Reviewer 1 Report
Comments and Suggestions for Authors
In this study, the authors used Cas9 induced VHL-mutation in the Caki-1 cell line model to investigate how VHL mutation followed by its restoration affects cellular heterogeneity in the context of renal cancer. As the main finding, they concluded that the restoration of VHL expression alone is not sufficient to fully revert the tumorigenic phenotype. The introduction presents sufficient background information on the field and justification for the study. The methods need a bit more information. Results are well described. Given the nature of the study, it would be very useful to look at the scripts for the downstream analysis after cell ranger. As it is an interesting work, I suggest the publication of the manuscript after minor changes, that I describe below:
Page 1, lines 26-27
Rephrase “Inactivating somatic mutations or deletions in the von Hippel-Lindau (VHL) 26 tumour suppressor gene region (chromosome arm 3p) are… ” as The inactivation of somatic mutations or deletions in the von Hippel-Lindau (VHL) tumor suppressor gene region (chromosome arm 3p) is… .
Page 3, line 98.
What is the cell ranger version? I would be great to say if multi, counts or aggr were used. Reference genome version needs to be cited as well.
Page 4, line 140.
What you mean by “The most DEGs”? The majority or the ones with the highest expression? It would be better to rephrase in a clearer way. This is the case for all “most” instances you use, such as the one in page 5, line 160.
Page 4, lines 146-147.
Maybe it’s worth rephrasing the sentence as “The identified DEGs between the two clusters are overlap with most of the ones reported in our prior bulk RNA-seq study”.
In the table 1 caption, replace "the most significantly enriched pathways" by "the enriched pathways with the highest significance".
Regarding the conclusion, what is the overall message in terms of what are the biological consequence of the persistence of pro-angiogenic cells across all genotypes? What is the contribution of the findings to the understanding of renal cancer progression and the role of VHL based on the gap in the field? Were all questions fully and successfully addressed?
As an overall suggestion, I think that short answers to those questions in the conclusion section would help the reader to have a clearer idea of the achievements of the study.
Comments on the Quality of English LanguageSome sentences need rephrasing to better explain what is intended, to make it clearer. I gave some suggestions on my report.
Reviewer 2 Report
Comments and Suggestions for Authors
In this study, Abramov et al aim to examine the phenotypes caused by the inactivation of VHL in clear cell renal cell carcinoma. Their approach of using a CRISPR/Cas9-induced VHL knockout cell line and comparing it to cells with reconstituted VHL is well thought out and has identified angiogenesis-associated genes as clinically relevant targets of VHL in kidney cancer.
Specific comments:
Please mention the source of the cell lines used in the study.
Please mention supplementary figure 1 in the results section as well to help readers find the details about the CRISPR-induced mutation.
Please change the terminology used to refer to the cells – as I understand there are 4 cell types used in the study:
- Parental Caki-1 cells – “WT”
- VHK CRISPR/Cas9 KO cells – “mutVHL”
- mutVHL cells transduced with WT VHL lentivirus – “HAVHL”
- mutVHL cells transduced with an empty lentivirus – “ctrl”
The name “control” for the last cell line is quite confusing to the reader. I recommend changing the names of HAVHL and ctrl to be something like “mutVHL-restored” and “mutVHL-control” to better indicate the origin of the cells.
Please comment on the level of expression of VHL in HAVHL cells. How does it compare to endogenous levels of VHL in WT Caki-1 cells?
Please specify whether HAVHL and ctrl cells were cloned with limiting dilution similar to mutVHL cells or if they are a polyclonal population.
The data in Fig 1 seem to indicate that the bulk datasets highly correlate with each other, as do the single cell datasets. However the correlation between each bulk dataset and its corresponding single cell dataset is quite poor. This is likely due to the lack of representation of low abundance transcripts in the single cell dataset (as the authors point out). The data in Fig 2 showing that cells with functional VHL cluster separately from the cells with mutant VHL are more convincing. I recommend the authors combine figures 1 and 2.
Please include a Venn diagram showing the overlap between DEGs in the bulk and single cell RNAseq datasets.
The VHL knockout cells were generated by isolating single cell clones. If the experiments were only done with a single clone, differences in gene expression between the parental cell line and the knockout (WT vs mutVHL) can arise either due to the VHL knockout or due to cloning from heterogeneity in the parental cell line. Please acknowledge this as a limitation of the study. Please include a table to address this – please show the pathways most enriched in mutVHL vs WT cells and then show their corresponding z-values.
Please show percentage of cells instead of cell counts in Fig. 3b.
Reviewer 3 Report
Comments and Suggestions for Authors
The authors focused their work on determining the significance of VHL gene inactivation in clear cell renal cell carcinoma using a targeted gene inactivation model with the CRISPR-Cas9 technique in the Caki-1 cancer cell line. Despite promising results and extensive work, critical errors in the project must be corrected. The Caki-1 cell line was derived from a ccRCC metastasis in the skin. These cells represent an advanced, mature form of ccRCC. However, a key aspect of this cell line is its VHL-independent model of carcinogenesis. VHL gene damage is observed in 70%-90% of ccRCC cases, as the authors of the study pointed out. The molecular phenotype of Caki-1 cells is based on the deregulation of pathways associated with carcinogenesis, such as PI3K-Akt-mTOR, Wnt, and TMEM, as well as increased integrin expression. Therefore, the deliberate inactivation of the VHL gene could be an interesting reference point for assessing carcinogenesis, but only under certain conditions. It is a critical mistake to rely on only one cell line without comparing it to others. The best model for many years has been 786-O or A-498, which have nonfunctional VHL proteins and overactivated HIF1A/HIF2A proteins as transcription factors. Had the authors conducted a preliminary comparison between Caki-1 (WT or VHL-) and 786-O lines, their work could have contributed to the identification of future molecular targets. However, the results of the study clearly show that VHL inactivation in late, mature, metastatic ccRCC and subsequent VHL restoration only partially restores the ratio between angiogenic and "normal" cells. These results confirm that tampering with the VHL gene is pointless in this Caki-1 cancer cell phenotype (as authors wrote in conclusions). Therefore, the authors should compare Caki-1 with 786-O and normal renal epithelial cells (e.g., HK2) for validation. Ideally, the Cas9 technique would be used to repair the VHL gene in the 786-O or A-498 cells.
Another critical element is the cell culture conditions. The authors should perform experiments in a normoxic environment, as has been done, and in a hypoxic environment, as has not yet been done, to study the conditions prevailing in the dominant ccRCC picture. In the case of hypoxia, pro-angiogenic cells would likely dominate, as in normoxia, "normal" Caki-1 cells have developed due to a comfortable environment, which differs greatly from the native one.
Finally, the authors should mention attempts to evaluate Caki-1 cell behavior with VHL disabled, e.g., migration and colony formation evaluation. Ideally, an animal model would be used, but that is a different topic. This is not a criticism, but rather an idea for future research.
The discussion is quite short and it does not show clear take-home message.
In conclusion, due to the flawed model of considering only the participation of the VHL gene in the mature, metastatic clear cell carcinoma Caki-1 line, I believe the paper does not meet the IJMS journal's high standards.
Round 2
Reviewer 2 Report
Comments and Suggestions for Authors
The authors have satisfactorily addressed all of my comments. However, the additions made to the supplement seem to be missing in the version sent to me for review. Since the authors have included the figures in their response to my review, I have reviewed the data and I am happy to support the publication of this manuscript. However, it is necessary to ensure that the figures are added to the supplementary materials before publication.
Author Response
Comments 1: The authors have satisfactorily addressed all of my comments. However, the additions made to the supplement seem to be missing in the version sent to me for review. Since the authors have included the figures in their response to my review, I have reviewed the data and I am happy to support the publication of this manuscript. However, it is necessary to ensure that the figures are added to the supplementary materials before publication.
Response 1: Thank you for your careful review and for bringing this oversight to our attention. We sincerely apologize for the missing supplementary files in the initial submission. We have now uploaded the complete set of supplementary materials to the submission portal. We appreciate your time and consideration and are pleased that you support the publication of our manuscript.